# Full-Attention Driven Graph Contrastive Learning: with Effective Mutual Information Insight

## ABSTRACT

Graph contrastive learning has shown significant promise in unsupervised scenarios. Many techniques endeavor to maximize the mutual information between two perturbed graphs, but challenges arise when the data augmentation alters the graph's informative attributes, leading to potential noise positive pairs. While recent approaches have tried addressing this issue, they have shortcomings in guaranteeing effective data augmentation or incurring high computational costs. Only a few researches try to do data augmentation in encoder's latent space. With the help of full-attention graph transformers we may get a wider encoder's latent space to do data augmentation, while using full-attention graph transformers still causes some problems like noise information. This paper introduces GACL (Graph Attention Contrastive Learning), a novel model that integrates the full-attention transformer with a message-passing-based graph neural network as the encoder. To avoid noise information with full-attention, GACL introduces a modification to the full-attention. Our GACL model uniquely addresses challenges posed by full-attention and offers an innovative data augmentation strategy. Finally, we establish the concept of effective mutual information and validate the effectiveness of full-attention data augmentation. Empirical evaluations confirm GACL's superior performance, cementing its position as a state-of-the-art(SOTA) solution in the field of graph contrastive learning. The anonymous code is available on https://anonymous.4open.science/r/GACLAnno-C424.

## CCS CONCEPTS

• **Computing methodologies** → **Learning latent representations**; **Unsupervised learning**; • **Information systems** → **Data mining**.

## KEYWORDS

Graph Representation Learning; Graph Contrastive Learning; Graph Transformer; Information Theory.

**ACM Reference Format:**
Anonymous Author(s). 2018. Full-Attention Driven Graph Contrastive Learning: with Effective Mutual Information Insight. In *Proceedings of Make sure to enter the correct conference title from your rights confirmation emai (Conference acronym 'XX)*. ACM, New York, NY, USA, 9 pages. https://doi.org/XXXXXXX.XXXXXXX

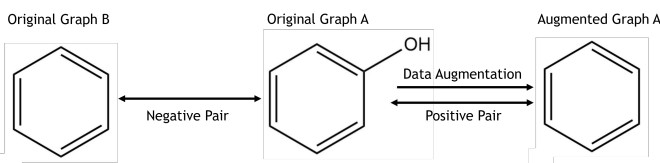

**Figure 1: This figure shows an example of ineffective positive pair. Graph A is phenol and Graph B is benzene. Obviously, Graph A and Graph B have completely different chemical properties and different labels. However, after data augmentation, we may augment the phenol to be benzene. We treat such kind of data augmentation as ineffective positive pair.**

## 1 INTRODUCTION

Graph supervised learning, which requires an extensive amount of labels and consequently, costing a significant labour effort. To address this challenge, graph contrastive learning [25], as a representative of self-supervised learning approaches [16, 35], has seen rapid development in recent years [24, 34, 40, 41], marking a new milestone in graph self-supervised tasks. This approach primarily draws inspiration from contrastive learning paradigms in computer vision [4, 10]. By utilizing data augmentation, it creates a new positive sample for each graph, thereby obtaining two perspectives of the graph, and then maximizes the mutual information between these perspectives to learn graph representations [15].

For instance, GraphCL [41] employs node dropping, edge perturbation, attribute masking or subgraph to perturb the original graph, resulting in two augmented versions of the graph. It then aims to maximize the mutual information shared between these two perturbed graphs. However, there are inherent issues with this methodology. Perturbing the original graph can potentially alter its informative attributes. As illustrated in Figure 1, although graph A has undergone data augmentation, it is evident that such augmentation may be ineffective or even detrimental. Consequently, the mutual information of this positive example can be perceived as noise. If the mutual information being maximized resembles the case shown in Figure 1, the effectiveness of contrastive learning would be significantly compromised. Several efforts have been initiated to tackle this issue. Some researches try to avoid the issue by manual trial [41] or automated search [40] or neighbourhood attributes guidance [24], while it's still hard to avoid such pairs. Some other researches[44] rely on domain-specific knowledge to avoid the issue, while introducing a considerable amount of prior knowledge, limiting its versatility. SimGRACE [34] creatively employs a perturbation encoder for a straightforward data augmentation scheme, avoiding such issues by augmenting data in encoder's latent space, while it's still lack of a theoretical analysis. We may introduce Graph Transformer to expand the encoder's latent space to achieve better performance and do some theoretical analysis for the latent space data augmentation.

On another note, these models often assume the perfection of encoders during their analysis [27], yet message-passing-based graph neural networks have their own set of challenges, such as over-smoothing and over-squashing [2, 28]. Thus, arises the question: can we enhance contrastive learning's performance by boosting the encoder's expressive capabilities? Graph Transformers [5, 13, 21, 32, 39], an emerging and promising model, offers a potential solution. Recently, a few studies have focused on contrastive learning around Graph Transformers. They use [7, 31, 42] sparse or hierarchical graph transformer for contrastive learning. However, these works have not adopted full-attention; they have incorporated the graph structure into attention to varying degrees, leading to the expansion of latent space not as much as expected. In fact, integrating full-attention in contrastive learning doesn't necessarily enhance the encoder's expressive capability, as in graph structures, the significance of neighbour nodes usually outweighs distant nodes. This typically necessitates a vast amount of graph labels. In label-deprived scenarios, full-attention itself might introduce ineffective positive sample pairs. We may need to transform the full-attention part to avoid its noise introduced by irrelevant nodes.

**Our solution.** To address the aforementioned challenges, we introduce our model, GACL(Graph Attention Contrastive Learning). Inspired by GraphGPS [20], we incorporate the full-attention transformer, which works in tandem with the message-passing-based graph neural network, serving as the encoder. In this way, we can get a wider encoder's latent space to do data augmentation in avoid of the ineffective positive samples on original graphs. Furthermore, by introducing a noise modification to the full-attention, we mitigate the potential of introducing ineffective positive pairs, thus enhancing the encoder's expressive capability. Concurrently, we utilize the noise-modified full-attention as a way of data augmentation. This approach offers a simple yet universal method of data augmentation while minimizing the likelihood of obtaining ineffective positive pairs. As depicted in Figure 1, our analysis revolves around the effectiveness of positive sample pairs. Thus, we introduce the concept of effective mutual information. The mutual information between positive sample pairs in Figure 1 is not considered effective mutual information; instead, it is deemed as ineffective mutual information. Building on this concept, we analyze the mutual information gains brought about by integrating the full-attention transformer, substantiating the ability of GACL to enhance model expressiveness. Simultaneously, viewing full-attention as a form of data augmentation, we validate the efficacy of full-attention data augmentation in contrastive learning by analyzing its reduction in ineffective mutual information, emphasizing its minimized introduction of noisy positive pairs.

Our study pioneers the integration of the full-attention transformer into contrastive learning. It addresses potential challenges posed by full-attention and positions it as a straightforward and effective data augmentation strategy. We delineate our contributions in three main points:

- We bolster the expressive capacity of the contrastive learning encoder by introducing the full-attention transformer. By leveraging the contrastive learning framework, we counteract the

effects imposed on the full-attention transformer in unlabeled contrastive learning scenarios.
- We present a solution to the issue of ineffective information in the full-attention transformer, advocating for its use as a simple yet effective data augmentation technique. Furthermore, we establish the concept of effective mutual information and, within its purview, validate the effectiveness of full-attention data augmentation. Our discourse on effective mutual information also highlights potential future directions to enhance graph contrastive learning performance.
- Extensive experiments over various datasets demonstrate the exceptional performance of our GACL model, surpassing other baseline models in its efficacy, achieving state-of-the-art (SOTA) performance.

## 2 RELATED WORK

### 2.1 Graph representation learning

Graph embedding techniques, as cited in references [19, 26], have proven their capability in deriving latent representations of a graph that maintain the graph's structure. However, in more recent developments, graph neural networks (GNNs) [12, 29, 37], highlighted in studies, have ascended as the leading methodologies for graph representation learning. Their rising prominence can be attributed to their adeptness at encapsulating both the individual node information and the overarching graph structures. This is achieved through an ingenious mechanism that amalgamates neighborhood data using a message-passing framework. For readers keen on delving deeper into the intricacies of GNNs, surveys are available in references [3, 33].

### 2.2 Graph contrastive learning

Contrastive learning has emerged as a powerful approach in representation learning by optimizing the similarity between augmented views of data. This paradigm has shown significant potential, especially in the context of graph data where the manner of data augmentation plays a pivotal role. Historically, unsupervised graph representation techniques, such as DeepWalk [19] and node2vec [6], have capitalized on a contrastive framework. These methods underscore the similarities derived from random walks on graphs and often utilize Noise-Contrastive Estimation (NCE) [8] to model the probabilities of node co-occurrence pairs. However, while these methods have laid the groundwork, they possess inherent limitations, including sensitivity to hyperparameter settings [6, 19].

In recent advancements, Graph Neural Networks (GNNs) have been employed as sophisticated encoders, leading to a myriad of novel approaches. For instance, DGI [30] integrates GNN with contrastive learning, emphasizing the maximization of mutual information [11] between global graph-level and local node-level embeddings. Taking cues from SimCLR [4], GraphCL [41] employs graph-level data augmentations but necessitates manual tuning, whereas JOAO [40] automates the augmentation selection process in GraphCL, albeit at a high computational cost.

A pertinent challenge lies in ensuring that random perturbations during data augmentation do not compromise the quality of the augmented views. MVGRL [9] offers an innovative approach,

advocating for the learning of both node- and graph-level representations. This is achieved by contrasting node representations with augmented graph summary representations following node diffusion. Several other methods, such as GRACE [43] and Infogcl [36], have been formulated to ensure the retention of critical graph information during augmentation. They leverage node centrality and information-aware representation models, respectively. GCA [44] endeavors to bolster semantic information preservation and finetune data augmentation techniques. Still, there remains the perennial challenge of designing adaptable and efficient data augmentations suitable for diverse graph structures. SimGRACE [34] presents a unique perspective by perturbing model parameters, thus obviating the need for input data augmentation altogether. ENGAGE [24] proposes a guided data augmentation using local embedding explanation.

With regrad to the recent contrastive learning on Graph Transformer, SGTC [42] integrates the sparse graph transformer for contrastive learning, while HEAL [7] employs a hierarchical graph transformer for protein function prediction, and TCL [31] introduces a graph transformer for dynamic graph modeling.

## 3 PRELIMINARIES

In this subsection, we introduce some preliminary concepts and notations that will be essential for our exposition throughout the paper.

**An attributed graph,** denoted by $G = (V, E)$, comprises:

- A set $V$ representing nodes. Each node $v \in V$ can have an attribute vector $X_v \in \mathbb{R}^F$, where $F$ denotes the dimensionality of the attribute space.
- A set $E \subseteq V \times V$ representing edges.

We often use a matrix $A$ to represent the adjacency matrix.

In many cases, the graph $G$ should also belong to a class, named label $Y$. If it does not have, we can assume it belonging to the latent class [22]. That means, each graph should have its label whether it's obvious or not.

**Information entropy** [23], often simply referred to as entropy, quantifies the amount of uncertainty or unpredictability associated with a random variable, typically a source of information.

In its most basic form, the entropy $H(X)$ of a discrete random variable $X$ with possible values $x_1, x_2, ..., x_n$ and corresponding probabilities $p(x_1), p(x_2), ..., p(x_n)$ is given by:

$$H(X) = -\sum_{i=1}^{n} p(x_i) \log p(x_i)$$

For two random variables $X$ and $Y$ with a joint distribution p(x, y), their joint entropy $H(X, Y)$ is defined as:

$$H(X, Y) = -\sum_{x \in X} \sum_{y \in Y} p(x, y) \log p(x, y)$$

Joint entropy quantifies the uncertainty or information content of the joint distribution of $X$ and $Y$. When we describe the common information of the two variables, we often use mutual information. For variables $X$ and $Y$, their mutual information $I(X; Y)$ is:

$$I(X; Y) = H(X) + H(Y) - H(X, Y)$$

## 4 METHODOLOGY

In this section, we delve deep into the intricacies of our model, GACL. An overview of the model's architecture is depicted in Figure 2. We will then focus on the aspects related to 'effective mutual information', providing a theoretical justification for the enhancements of our GACL model and the efficacy of its data augmentation technique. Guided by the theory of effective mutual information, we will subsequently explore potential future developments in the contrastive learning loss function.

### 4.1 GACL

As illustrated in Figure 2, in comparison to typical contrastive learning encoders, our model prominently features the integration of the full-attention transformer module. Drawing inspiration from GraphGPS [20], we have combined the full-attention transformer with the message-passing graph neural network, but we do not use any positional encoding for the sake of fairness compared to other models. The adjacency matrix information of the graph is exclusively utilized in the message-passing graph neural network, while the full-attention module, aimed at enhancing the encoder's capability, is fed only with node encoding information. However, it is important to note that the mere integration of the full-attention transformer module into the encoder does not necessarily guarantee an enhancement in the model's expressive capacity, as supported by experiments discussed in section 5.3. Given that nodes in a graph typically rely more on their neighboring nodes, there aren't always a significant number of effective long-range node connections. What truly influences the effectiveness of full-attention is the proportion of useful information within the attention details.

In light of this, we introduce a noise matrix into the full-attention mechanism. The elements of the noise matrix are composed by {0, 1} or {0, -1} or {1, -1}. The size of the noise matrix is the same as attention. Since it's challenging to pinpoint which attention weights are genuinely effective, the noise matrix adopts a random distribution, formulated as:

$$NoiseAttention(Q, K, V) = [\mathbf{N} \cdot softmax(\frac{QK^T}{\sqrt{d_k}})]V, \qquad (1)$$

where $\mathbf{N}$ is the noise matrix, and $Q, K, V$ is query, key and value of the transformer. The attention matrix undergoes element-wise multiplication with the noise matrix. As an example, with the noise matrix composed by {0, 1}, each element has possibility $p$ to be 1.

We refer to the representation processed through the noise matrix as the positive sample. This representation, in conjunction with the representation that has not undergone noise matrix processing, forms a positive pair, thereby fulfilling the data augmentation requirement in contrastive learning. It is imperative to note that we do not undertake any structural modifications on the original graph, so we avoid semantic changes caused by structural modifications. Instead, our method focuses on processing the attention with the noise matrix. To provide a more intuitive understanding, consider the analogy of applying data augmentations to the primary graph. Within the framework of the full-attention matrix, there's an inherent presumption that all nodes are interconnected, depicting a fully-connected graph. Introducing noise into this attention matrix

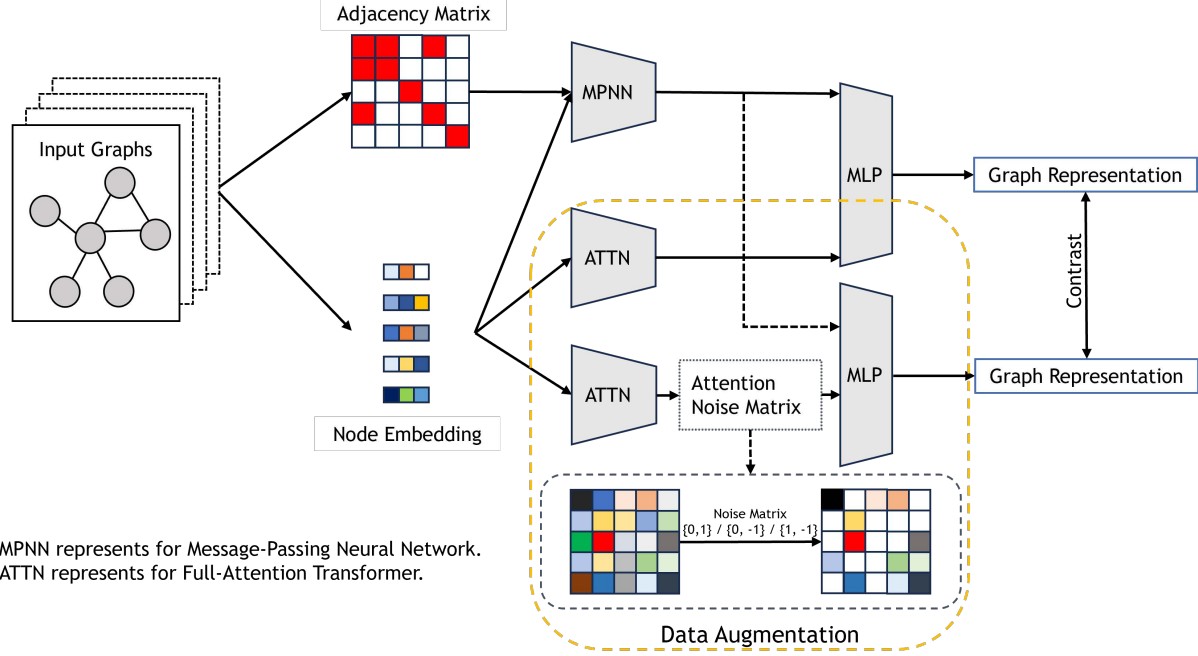

**Figure 2: This figure shows our model GACL. In our model, we use MPNN like GCN or GIN to encode the adjacency matrix information. We add a full-attention part which will involve unnecessary information. To both enhance the encoder and work as the data augmentation part, we propose an attention noise matrix and explain why it works theoretically.**

can be likened to edge permutations within such a fully-connected graph.

For our contrastive learning loss, we employ the InfoNCE loss, which plays a pivotal role in augmenting the proportion of effective information within the attention. We will talk about it in section 4.3. In Section 4.2, we will formally introduce the definition of effective mutual information along with its associated formulas, and Section 4.3 will be dedicated to a detailed analysis of the efficacy and roles of the various modules within our model.

This elaboration aims to present the model's techniques and rationale, making it appropriate for inclusion in a scholarly article.

## 4.2 Effective mutual information

As previously discussed, effective mutual information plays a pivotal role in contrastive learning. In this subsection, we will delve into the concept of effective mutual information mentioned earlier and provide its formal definition. First, we present the definition of effective mutual events.

DEFINITION 1. *(effective events) In the realm of contrastive learning, the goal is to encapsulate information approaching $I(G; Y)$. Given an event set $E$ with associated information entropy $H(E)$, where $H(E) = I(G; Y)$, If event $x \in E$, it is deemed an effective event, and $E$ is named an effective event set.*

In this definition, $H(E) = I(G; Y)$ means that, for any subset $S \subseteq E$, $H(S) < I(G, Y)$ holds true. Conversely, for any super-set $E \subseteq S$, $H(S) > I(G, Y)$ holds true, too.

This definition tells us when the information of the event is useful in fact. Here the set $E$ may be abstract now. In fact, it comes along with the actual label and the distribution of the whole dataset. The same as the definition of effective events, we can give a definition of the effective entropy.

DEFINITION 2. *(effective entropy) Given an event $s \in S$ and an effective event set $E$, the entropy of events in the intersection $s \in S \cap E$ is termed the effective entropy $H_e(S)$. We have similar definition for the the ineffective entropy $H_n(S)$.*

Contrastive learning learned from two differing views. The view can also be seen as an event. We present the definition of effective mutual information.

DEFINITION 3. *(effective mutual information) Let $u$ and $v$ denote two views. $u$ belongs to a view set $U$ and $v$ belongs to $V$. $E$ is an effective event set. Only when both $u \in E$ and $v \in E$ are effective views, this part of the mutual information of $U$ and $V$ is termed the effective mutual information, $I_e(U; V)$, formulated as:*

$$I_e(U; V) = H_e(U) + H_e(V) - H_e(U; V), \qquad (2)$$

$$I_e(U; V) = \sum_{v \in V \cap E} \sum_{u \in U \cap E} p(u, v) log \frac{p(u, v)}{p(v)p(u)}. \qquad (3)$$

All other mutual information contributions are classified as 'ineffective' or 'noise', denoted $I_n(U; V)$. From this definition, we can understand the two parts of the mutual information in contrastive learning, the effective mutual information $I_e$ and the ineffective mutual information $I_n$. That is to say,

$$I(U; V) = I_e(U; V) + I_n(U; V) \qquad (4)$$

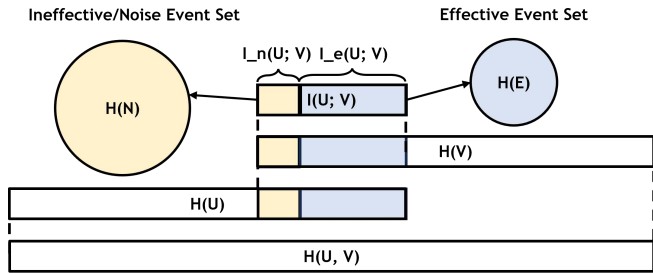

**Figure 3: This figure shows the relationship between two views $U$, $V$ and their mutual information, effective and ineffective mutual information.**

We describe it vividly in figure 3.

Correspondingly, the ineffective mutual information splits into three components:

$$I_n(U;V) = \sum_{v \in V-E} \sum_{u \in U \cap E} p(u,v) log \frac{p(u,v)}{p(v)p(u)}$$
$$+ \sum_{v \in V \cap E} \sum_{u \in U-E} p(u,v) log \frac{p(u,v)}{p(v)p(u)}$$
$$+ \sum_{v \in V-E} \sum_{u \in U-E} p(u,v) log \frac{p(u,v)}{p(v)p(u)} \quad (5)$$

As we defined before, the effective mutual information is which actually works in the mutual information between the two views. In fact, what truly affects the final outcomes of views $u$ and $v$ is whether the effective mutual information $I_e(U;V)$ is sufficient and whether the ineffective mutual information $I_n(U;V)$ is minimized. Intuitively, we can consider the effective mutual information as the correlation between the graph and the labels, while the ineffective mutual information represents some shared noise information between the views.

## 4.3 Analysis of GACL

In this subsection, drawing upon the theoretical framework presented in Section 4.2, we will analyze the effectiveness of the various components of the GACL model introduced in Section 4.1. We will analyze it in three parts soon.

Firstly, we will prove that combining full-attention transformer does increase the mutual information between two views.

THEOREM 1. *For contrastive learning, combining with full-attention transformer will improve the expressive ability of the message-passing graph neural network, if the noise of the full-attention transformer is low.*

PROOF. For the original graph $G$, after data augmentation, we will get two views $G_u$ and $G_v$. Use $M()$ to denote the message-passing part of the network, $T()$ to denote the full-attention transformer part of the network. When combining the two encoders together, we have:

$$I(U;V) = I(M(G_u), T(G_u); M(G_v), T(G_v))$$
$$= I(M(G_u), M(G_v)) + I(M(G_u); T(G_v)|M(G_v)) +$$
$$I(M(G_v); T(G_u)|M(G_u)) + I(T(G_u); T(G_v)|M(G_u), M(G_v))$$
$$\geq I(M(G_u), M(G_v)). \quad (6)$$

The above formula holds since the mutual information should always be greater or equal than zero. To dissect the sources of this accrued mutual information, we identify three primary contributors: $I(M(G_u); T(G_v)|M(G_v))$, its symmetric counterpart $I(M(G_v); T(G_u)|M(G_u))$, and $I(T(G_u); T(G_v)|M(G_u), M(G_v))$. Given the omission of the positional encoding and the total different latent representation between $M()$ and $T()$, the primary contribution to mutual information arises from the attention mechanisms in $T()$, that is $I(T(G_u); T(G_v)|M(G_u), M(G_v))$.

If the noise of the full-attention transformer is low, that means $\frac{H_e(T(G))}{H(T(G))}$ is high, thus $\frac{I_e(T(G_u);T(G_v))}{I(T(G_u);T(G_v))}$ is high. So the growing part of the mutual information helps improve the expressive ability of the message-passing graph neural network by increasing the effective mutual information between the two views.

□

From this theorem, we know that the key point is to denoise the full-attention part. We use attention noise matrix in our GACL. In GACL, the representation with attention noise matrix is the positive pair with the original representation. We consider their mutual information.

THEOREM 2. *The attention noise matrix {0,1} will decrease the ineffective mutual information brought by the full-attention part in graph contrastive learning.*

PROOF. To simplify the representation, we use $A$ to denote the full-attention part and $p$ to denote the possibility of each element in attention noise matrix to be 1. We have:

$$I_n(A; \mathbf{N} \cdot A)$$
$$= H_n(A) + H_n(\mathbf{N} \cdot A) - H_n(A, \mathbf{N} \cdot A)$$
$$= H_n(A) + H_n(\mathbf{N} \cdot A) - H_n(A) - H_n(\mathbf{N} \cdot A|A)$$
$$= H_n(\mathbf{N} \cdot A) - H_n(\mathbf{N} \cdot A|A)$$
$$= -(1-p) \times \log_2(1-p) + p \times H_n(A) - H_n(\mathbf{N})$$
$$= p \times H_n(A) + (1-p) \times \log_2(\frac{1}{1-p}) - 1$$
$$< H_n(A). \quad (7)$$

Since $H_n(A)$ is the original mutual information brought by the full-attention part, the ineffective mutual information between the two views decreases. □

In this theorem, we get to know that the ineffective mutual information between two views decreases by the attention noise matrix. However, the effective mutual information get decreased at the same time.

In our GACL, we use InfoNCE loss to recover the effective mutual information brought by the full-attention part.

THEOREM 3. *InfoNCE loss helps recover the effective mutual information with attention noise matrix brought by the full-attention part.*

Proof. Since no modifications are done to the original graph $G$, $G_u$ is $G_v$ in fact. Denoting $\mathbf{N} \cdot T(G)$ as $T'(G)$, we have:

$$I_e(T(G_u); T(G_v)|M(G_u), M(G_v))$$
$$=I_e(T(G); \mathbf{N} \cdot T(G)|M(G))$$
$$=H_e(T(G)|M(G)) - H_e(T(G)|T'(G), M(G)). \tag{8}$$

Next we will omit $G$ for convenience. Since $H_e(T(G)|M(G))$ doesn't have any changes, and

$$H_e(T|T', M)$$
$$= \sum_{t' \in T' \cap E} \sum_{m \in M \cap E} P(t', m) H(T|T' = t', M = m)$$
$$= -\sum_{t' \in T' \cap E} \sum_{m \in M \cap E} P(t', m) \sum_{t \in T \cap E} P(t|t', m) \log P(t|t', m), \tag{9}$$

thus recovering the effective mutual information equals to increasing the probability of $P(t|t', m)$.

Fundamentally, the InfoNCE loss facilitates the estimation of the lower bound of mutual information concerning the positive pairs. Let's consider $U$ and $V$ as positive views. The essence of what the InfoNCE loss optimizes can be depicted as:

$$P(u|v) = \frac{\exp(s(u, v))}{\sum_{u \in U} \exp(s(u, v))}. \tag{10}$$

Through optimizing $P(u|v)$, we optimize $P(t, m|t', m)$, that is $P(t|t', m)$. With this assist from contrastive loss, $I_e(T; T'|M)$ undergoes optimization. Given that $I(T; T'|M)$ is constrained by the probability $p$ intrinsic to the noise matrix, the mutual information ratio of $I(T; T'|M)$ is set on an upward trajectory with the incorporation of contrastive learning. That is to say, infoNCE loss helps recover the effective mutual information with attention noise matrix brought by the full-attention part.

□

From the proof, we can conclude that the possibility $p$ of the attention matrix, which reduces the ineffective mutual information of the full-attention, also limits the upper bound of the effective mutual information.

## 4.4 Further discussion about effective mutual information

Proceeding from section 4.3, it's imperative to engage in a deeper exploration of the nexus between contrastive loss and effective mutual information. Within the contours of contrastive learning, it's axiomatic that the two views ought to form positive pairs. Yet, a singular focus on positive pairs obscures our understanding of what truly encapsulates 'effectiveness', primarily because the term 'effective' often orbits around tangible or latent labels. This brings us to the observation that in contrastive learning, negative pairs are typically characterized by divergent tangible or latent labels. Through this lens, effective mutual information invariably has ties not just to the positive pairs but spans its influence to the negative pairs as well. This leads us to the following proposition:

Proposition 4. *For the ineffective mutual information $I_n(U; V)$, it's all the union of all the mutual information among the original one, the positive one and the negatives. That is to say,*

$$I_n(U; V) = \lim_{card(N) \to +\infty} \forall i \in N, \sum I(U; V_{pos}; V_i) \tag{11}$$

*N refers to the set of negative samples.*

From Definition 3.1, it's established that effective events contribute to the information entropy present in the mutual information between the graph and the label, $I(G; Y)$. This implies that the latent label of a graph $G$ is distinct. If another graph does not have the same label as $G$, it shouldn't share equivalent mutual information with $G$'s positive pair. Thus, $I(U; V_{pos}; V_i)$where $i \in N$) would be a fragment of ineffective mutual information and wouldn't contribute to $I(G; Y)$.

As the set of negative pairs approaches infinity, it provides an extensive ineffective set of negative pairs, ensuring ineffective mutual information is accounted for. The residual mutual information between $I(U; V)$ then becomes $I_e(U; V)$. Alternatively viewed, if effective events feature in $I_e(U; V)$, the label's information is distinctive enough to exclude other contrasting examples.

With this elucidation, the approach to gauge the effective mutual information between positive pairs is now clearer, leading to a refined objective for contrastive learning. First, it's crucial to augment the mutual information shared among positive pairs. Secondly, the effective portion of this shared mutual information should dominate, thereby sidelining the ineffective information unrelated to the latent label.

Yet, the estimation of $I(U; V_{pos}; V_{neg})$ remains an intricate puzzle. To offer a perspective:

$$I(U; V_{pos}; V_{neg}) = I(U; V_{pos}) - I(U; V_{pos}|V_{neg}) \tag{12}$$

$$I(U; V_{pos}|V_{neg}) = \sum_{u, v_p, v_n} p(u, v_p, v_n) \log \frac{p(v_n)p(u, v_p|v_n)}{p(u|v_n)p(v_p|v_n)} \tag{13}$$

To curtail $I(U; V_{pos}; V_{neg})$, the term $I(U; V_{pos}|V_{neg})$ must be maximized, particularly the ratio $\frac{p(v_n)p(u, v_p|v_n)}{p(u|v_n)p(v_p|v_n)}$. Unlike the straightforward cosine similarity between two variables, quantifying the intersection of three elements is a convoluted endeavor warranting further exploration.

We can also employ mutual information between two elements to approximate $I(U; V_{pos}; V_{neg})$. As stated, the aim is to diminish this. Evidently:

$$I(U; V_{pos}; V_{neg}) \leq I(U; V_{neg}) + I(V_{pos}; V_{neg}) \tag{14}$$

Thus, to minimize $I(U; V_{pos}; V_{neg})$, we can target the upper bound $I(U; V_{neg}) + I(V_{pos}; V_{neg})$.

Incorporating the foundational goal of maximizing $I(U; V_{pos})$, an optimal strategy would be to magnify $I(U; V_{pos}) - I(U; V_{neg}) - I(V_{pos}; V_{neg})$. This approach synergizes the dual objectives of contrastive learning: enhancing both the mutual and effective mutual information amid positive pairs.

## 5 EXPERIMENTS

In this section, our primary aim is to evaluate the performance of our model, GACL.

## 5.1 Experimental Setup

**Datasets.** We utilize six datasets from the benchmark TUDataset [17] for our experiments. These datasets encompass graph data

| Input | Method | MUTAG | NCI1 | PTC-MR | PROTEINS | COLLAB | IMDB-B | Rank($\downarrow$) |
|---|---|---|---|---|---|---|---|---|
| A,X,Y | GCN | 85.6 ± 5.8 | 80.2 ± 2.0 | - | 74.9 ± 3.3 | 79.0 ± 1.8 | 70.4 ± 3.4 | |
| | GIN | 89.4 ± 5.6 | 82.7 ± 1.7 | - | 76.2 ± 2.8 | 80.2 ± 1.9 | 75.1 ± 5.1 | |
| A,X | node2vec | 72.6 ± 10.2 | 54.9 ± 1.6 | 58.6 ± 8.0 | 57.5 ± 3.6 | 56.1 ± 0.2 | 50.2 ± 0.9 | 9.3 |
| | sub2vec | 61.1 ± 15.9 | 52.8 ± 1.5 | 60.0 ± 6.4 | 53.0 ± 5.6 | - | 55.3 ± 1.5 | 9.6 |
| | graph2vec | 83.2 ± 9.3 | 73.2 ± 1.8 | 60.2 ± 6.9 | 73.3 ± 2.1 | - | 71.1 ± 0.5 | 7.6 |
| | InfoGraph | 89.0 ± 1.1 | 76.2 ± 1.0 | 61.7 ± 1.4 | 74.4 ± 0.3 | 70.7 ± 1.1 | 73.0 ± 0.9 | 4.7 |
| | GraphCL | 86.8 ± 1.3 | 77.9 ± 0.4 | 61.3 ± 2.1 | 74.4 ± 0.5 | 71.4 ± 1.1 | 71.1 ± 0.4 | 6.2 |
| | JOAO | 87.4 ± 1.0 | 78.1 ± 0.5 | - | 74.6 ± 0.4 | 69.5 ± 0.4 | 70.2 ± 3.1 | 7 |
| | AutoGCL | 88.6 ± 1.1 | 82.0 ± 0.3 | - | **75.8 ± 0.4** | 70.1 ± 0.7 | **73.3 ± 0.4** | 3.6 |
| | iGCL | 89.8 ± 1.2 | 82.7 ± 0.4 | - | 74.8 ± 0.5 | 72.0 ± 0.8 | 72.6 ± 0.6 | 3.2 |
| | SimGRACE | 89.0 ± 1.3 | 79.1 ± 0.4 | - | 75.4 ± 0.1 | 71.7 ± 0.8 | 71.3 ± 0.7 | 4.2 |
| | EG-simCLR | - | 83.0 ± 0.2 | 61.5 ± 2.4 | 75.4 ± 0.7 | 76.6 ± 1.3 | 71.8 ± 1.3 | 3 |
| | GACL | **90.1 ± 0.8** | **83.3 ± 0.4** | **62.6 ± 1.9** | 75.5 ± 0.7 | **77.8 ± 0.8** | 72.6 ± 1.0 | **1.5** |

**Table 1: Results for graph classification tasks. Our GACL model consistently demonstrates superior or comparable performance across various datasets relative to other cutting-edge methods. This underscores GACL's robustness and efficiency. To derive the positive pair from the anchor point, instead of using original data augmentation, we multiply the attention matrix with a noise matrix.**

sourced from diverse domains. Specifically, MUTAG, NCI1, PTC-MR, and PROTEINS are associated with small molecules and bioinformatics, whereas COLLAB and IMDB-BINARY pertain to social networks. Detailed specifications of these datasets can be found in Table 1.

| | Graphs | Classes | Avg. Nodes | Avg. Edges | Data Source |
|---|---|---|---|---|---|
| MUTAG | 188 | 2 | 17.93 | 19.79 | Small molecules |
| NCI1 | 4110 | 2 | 29.87 | 32.3 | Small molecules |
| PTC-MR | 344 | 2 | 14.29 | 14.69 | Small molecules |
| PROTEINS | 1113 | 2 | 39.06 | 72.82 | Bioinformatics |
| COLLAB | 5000 | 3 | 74.49 | 2457.78 | Social networks |
| IMDB-BINARY | 1000 | 2 | 19.77 | 96.53 | Social networks |

**Table 2: Summary of datasets**

**Evalution Protocols.** To evaluate the efficacy of our GACL model's graph-level unsupervised learning representations, we adopted protocols established by prior research [25, 41]. The model is trained using the entire dataset to acquire graph representations, which are subsequently fed into a downstream SVM classifier for 10-fold cross-validation. Evaluation is conducted every 10 epochs, with the best evaluation epoch chosen as the final result, aligning our method with the [25] approach. Each experiment is reiterated ten times, with the average being reported as the final score.

**Compared Baselines.** Our GACL model is benchmarked against various methodologies, encompassing supervised methods such as GCN and GIN. Additionally, we have incorporated comparisons with state-of-the-art graph self-supervised learning techniques, including node2vec [6], sub2vec [1], graph2vec [18], Infograph [25], GraphCL [41], JOAO [40], AutoGCL [38], iGCL [14], simGRACE [34], and EG-simCLR [24].

**Training environment.** All experiments were executed on a server equipped with an Intel(R) Xeon(R) Platinum 8358P CPU, an A40 48GB GPU, and 80 GB of RAM. It should be noted that results might exhibit minor variations across different training environments.

## 5.2 Performance evaluation

Graph classification tasks were performed on the aforementioned six benchmark datasets. For GACL, GCN [12] was employed as the message-passing neural network encoder. In terms of the transformer encoder, we opted for the full-attention mechanism and refrained from employing any positional or structural encoding during the encoding of the original graph data. The outcomes are illustrated in Table 2. Scores denoted by '-' indicate unavailability in their respective publications. Our GACL model's average rank stands at 1.5, outperforming all other methods. For the majority of datasets, our model surpassed the performance of leading-edge methodologies, highlighting the robustness and versatility of GACL. Importantly, our data augmentation process preserves the integrity of the original graph, thus ensuring the generalizability of GACL.

## 5.3 Analysis of the Attention Noise Matrix

In assessing the efficacy of our attention noise matrix, we contrasted its performance with GraphCL paired with a full-attention transformer. GraphCL primarily employs techniques like node dropping, edge permutation, and subgraphing for data augmentation. We retain its data augmentation strategy and integrate it with a full-attention transformer in a manner akin to our GACL encoder. The results of this comparison are depicted in Figure 5.

Figure 4 illustrates that the mere addition of a full-attention transformer to the message-passing graph neural network does not markedly enhance performance. Notably, performance metrics for the PTC-MR and PROTEINS datasets even decline due to the incorporation of noisy mutual information in the full-attention matrix. Our GACL model, augmented by the attention noise matrix, consistently surpasses traditional data augmentation, irrespective of whether the encoder solely uses the message-passing graph neural network or is paired with a full-attention transformer.

In another experiment centered on the noise matrix in NCI1, we initially varied the probabilities $p$ of the noise matrix to assess its influence. Subsequently, we incorporated alternative noise matrices

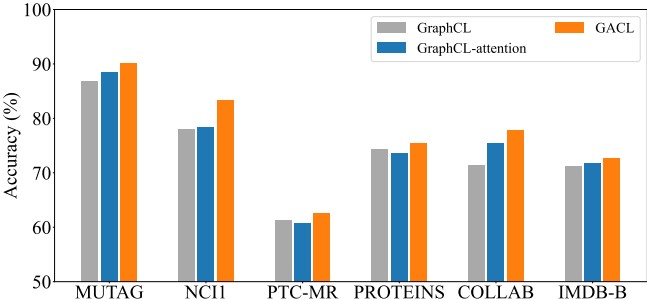

Figure 4: Comparison of performance between GraphCL, GraphCL equipped with full-attention transformer, and our GACL model.

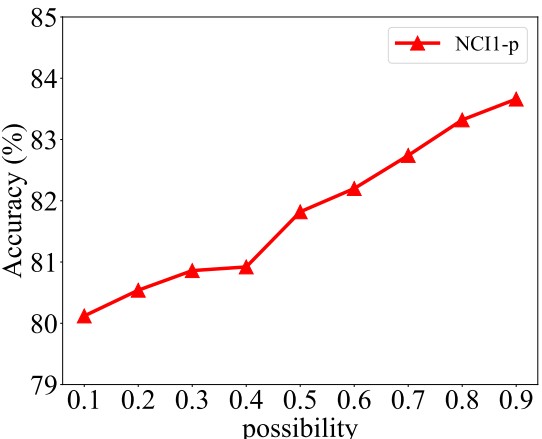

Figure 5: Performance variations on the NCI1 dataset as a function of the increasing probability $p$ associated with the attention noise matrix.

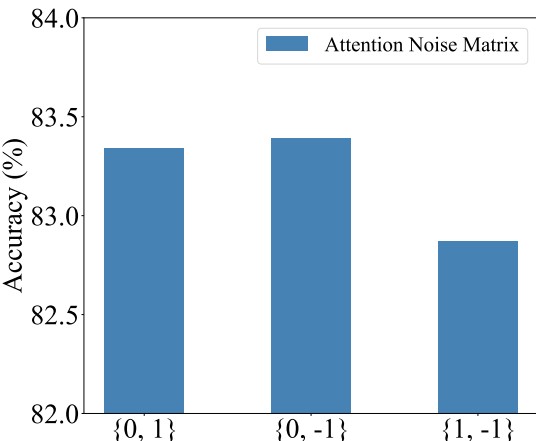

Figure 6: Performance variations on the NCI1 dataset when using different attention noise matrix.

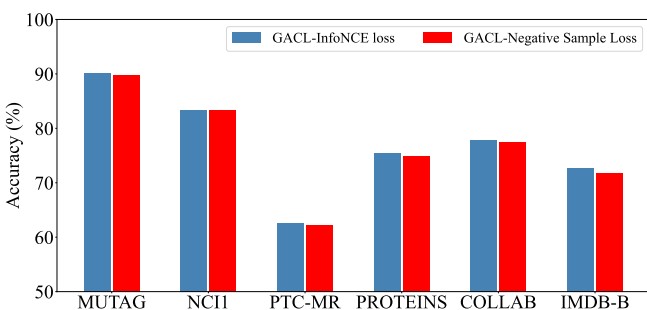

Figure 7: Performance when using different contrastive loss.

### 5.4 Negative sample based loss

As we mentioned is section 4.4, according to the formula 12, we simply conduct an experiment about the loss $I(U; V_{pos}) - I(U; V_{neg}) - I(V_{pos}; V_{neg})$. The results are shown in Figure 7.

From this figure, we can see that simply minimize the upper bound of $I(U; V_{pos}; V_{neg})$ may not help a lot in most datasets. We need to find a more concrete way to minimize $I(U; V_{pos}; V_{neg})$. That's our future direction.

## 6 CONCLUSIONS

In this paper, we introduce that our GACL model integrates a full-attention transformer into the encoder component of graph contrastive learning, leading to the expansion of encoder's latent space effectively. We introduce a theoretical framework grounded in effective mutual information. Under the guidance of the theoretical framework, we address the noise information introduced by full-attention. Concurrently, we employ the full-attention component as a universal data augmentation strategy in avoid of the complex and ineffective data augmentation to the graph structure. Empirically, GACL surpasses other baseline models, achieving state-of-the-art(SOTA) performance. At last, we derive insights for the possible direction of enhancing the efficiency of graph contrastive learning.

like -1,0 and -1,1 for a comparative analysis. The outcomes of this exercise are presented in Figure 6 and Table 7.

Figure 5 clearly demonstrates that as the probability linked with the attention noise matrix escalates, there's a corresponding improvement in performance. This is especially evident when the probability $p$ ranges between 0.4 and 0.5. For the NCI1 dataset, given its inherent data sparsity, the noise within the full-attention matrix might be pronounced, necessitating a higher probability for the attention noise matrix to effectively negate the effects of the noisy mutual information.

From Figure 6, it's evident that certain noise matrix, like {-1, 0}, exert minimal influence on performance by reducing the noise mutual information, even there is a few performance increase. The impact of the {-1, 1} noise matrix on noise mutual information is a little difficult and uncertain, resulting slight decrease in performance. It can not denoise the attention matrix well.

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
