# OpenReview forum: "Full-Attention Driven Graph Contrastive Learning: with Effective Mutual Information Insight"
_ACM.org/TheWebConf/2024/Conference — TheWebConf24_

### Official Review · Reviewer_Vz1e · 2023-11-23

**Novelty:** 5
**Technical Quality:** 4

**Review:**

This paper proposes to use 'full-attention graph transformer' to help generate effective contrastive pairs. The method can provide a wider encoder's latent space while avoiding problems, such as noisy information and ineffective augmentations. The solution utilizes a 'noise matrix' multiplied by the full-attention matrix. The author also provides the definitions of 'effective mutual information' and several theoretical results to support the method.

Pro:
1. The graph transformer is a relatively new subject to graph algorithms and thus has good potential in the field.
2. Using transformer to improve contrastive pairs is novel.

Cons:
1. The paper uses many confusing notations, making it hard to read. For example:
     (a). E and V denote the edge and node sets on page 3, but are used again to denote event and view on page 4.
2. The paper is implicit on how to construct 'noise matrix' and 'effective event set', especially since there is no provided pseudocode. On page 3, the author states: 'The elements of the noise matrix are composed by {0, 1} or {0, -1} or {1, -1}' but gives little explanation on what they mean and how to assign a 'pair' into the element of a matrix. Do you sample each element from the pair? The use of 'noise' in this paper is also confusing. From what I understand, 'noise' refers to 'ineffective contrastive pairs'. The author does not explain how the 'noise matrix' can avoid the issue, given that most contrastive learning methods are 'self-supervised'.
3. The theoretical parts are also confusing to me. Definition 1 defines 'effective event' by 𝐻(𝐸) = 𝐼(𝐺;𝑌). The author does not explain what 'event' corresponds to in a graph or a transformer. In definition 3, the author states '𝑢 ∈ 𝐸 and 𝑣 ∈ 𝐸 are effective views'. From what I understand, the 'event' refers to a (sub-)view, but I'm really confused about how to apply definition 1 on views to determine whether it's 'effective' or not.
4. The author does not use any 'positional encoding for the sake of fairness compared to other models'. I'm not convinced by the statement. Positional encoding has proved to be a vital component in transformers, which is also stated in [20].

**Questions:**

1. How are the 'noise matrix' and 'effective event set' obtained? Those two terms are fundamental to your paper.
2. How do you decide which views are effective without knowing the downstream task? In my opinion, any pair of views can be effective or ineffective in a self-supervised setting. The effectiveness heavily depends on the specific downstream task which is unknown. The definition provided '𝐻(𝐸) = 𝐼(𝐺;𝑌)' is confusing to me.

**Ethics Review Description:**

N.A.

**Reviewer Confidence:**

3: The reviewer is confident but not certain that the evaluation is correct

**Scope:**

3: The work is somewhat relevant to the Web and to the track, and is of narrow interest to a sub-community

---

### Official Review · Reviewer_tUK8 · 2023-11-23

**Novelty:** 4
**Technical Quality:** 6

**Review:**

Quality

Exhibits a good level of technical depth, but some aspects could benefit from clearer explanations. The experiments, while comprehensive, lack a broader range of datasets for more robust validation.

Clarity

Generally well-organized, but the complexity of the content sometimes hampers clarity, especially in dense theoretical sections.

Originality

Introduces some novel elements, like the noise matrix, but these concepts don't markedly distinguish the work from existing research in the field.

Significance

Makes a modest contribution to addressing challenges in graph contrastive learning, with potential implications that are yet to be fully realized in practical applications.


Pros

Innovative Approach: The integration of full-attention transformers with graph neural networks represents a significant advancement in the field.
Effective Solution to Existing Problems: The GACL model effectively addresses noise and inefficiency issues in graph contrastive learning.
Empirical Validation: Extensive experiments demonstrate the model's superior performance compared to other methods.
Theoretical Contributions: Introduction of concepts like effective mutual information adds theoretical depth.

Cons

Complexity: Some theoretical aspects may be overly complex for readers without a deep background in the field.
Potential for Overfitting: While not explicitly discussed, the complexity of the model might pose risks of overfitting, especially in smaller datasets.
Generalizability: The application of the model to a wider range of real-world scenarios remains to be explored.
Computational Demand: Although more efficient than some existing methods, the full-attention mechanism may still demand considerable computational resources.

**Questions:**

Methodological Clarification: Could you provide further details on how the noise matrix in the full-attention mechanism specifically contributes to reducing noise and improving data augmentation? Clarification on this could better highlight the novel aspects of your approach.

Theoretical Complexity: Some sections of the paper, particularly those dealing with theoretical foundations, are quite complex. Can you provide a more simplified explanation or intuitive understanding of these concepts? This could help in assessing the accessibility and clarity of your work.

Comparison with Existing Methods: While you mention several existing methods, a more detailed comparison with these methods, especially in terms of computational efficiency and performance, would be beneficial. How does GACL compare in terms of resource demands and scalability?

Overfitting Concerns: Could you address potential overfitting issues, especially in scenarios involving smaller datasets? Your response could help evaluate the robustness of the GACL model.

**Ethics Review Description:**

No Ethics problem

**Reviewer Confidence:**

3: The reviewer is confident but not certain that the evaluation is correct

**Scope:**

3: The work is somewhat relevant to the Web and to the track, and is of narrow interest to a sub-community

---

### Official Review · Reviewer_RZV1 · 2023-11-27

**Novelty:** 4
**Technical Quality:** 3

**Review:**

This work integrates a full-attention transformer into the encoder component of graph contrastive learning.

Pros:

1）	This work has established the concept of effective mutual information.

2）	Theoretical analysis is provided.

Cons:

1）	This work is not well presented. The proposed method is mainly introduced in Sec. 4.1, but not even complete and mathematical descriptions of the method are provided. The framework in Figure 2 also lacks clarity. So many details are unclear for readers.

2）	In abstract/introduction/conclusion, the authors mentioned the expansion of latent space as a key benefit of the proposed method. However, the methodology section does not provide any explanation on this point.

3）	The performance gains achieved by the proposed method is neglectable. For example, on MUTAG and NCI1, the gap over the second-best baselines is only 0.3, which is even smaller than the standard deviations. Moreover, it does not exhibit consistent superiority over existing methods. Thus, the effectiveness of proposed method is less satisfactory.

4）	The introduction of full-attention should result in additional computations with quadratic complexity, but relevant analysis (theoretical or experimental) is not provided.

5）	Many indexes in some texts do not match the numbering of the figures and tables.

**Questions:**

Please refer to cons above.

**Reviewer Confidence:**

3: The reviewer is confident but not certain that the evaluation is correct

**Scope:**

3: The work is somewhat relevant to the Web and to the track, and is of narrow interest to a sub-community

---

### Official Review · Reviewer_6Nyj · 2023-11-27

**Novelty:** 4
**Technical Quality:** 4

**Review:**

# Summary:

To address the challenge of noise information induced by the full-attention transformer, this paper proposes a novel approach, Graph Attention Contrastive Learning (GACL). This paper combines the full-attention transformer with message-passing-based graph neural networks and modifies the full-attention transformer by introducing a noise matrix into the mechanism. Furthermore, this paper introduces the concept of effective mutual information and validates the effectiveness of full-attention data augmentation within its scope. The issues proposed in this paper have certain research value. Extensive experiments on various datasets demonstrate the outstanding performance of the GACL model, surpassing other baseline models in terms of efficacy, highlighting the robustness and versatility of GACL.

# Strong points:

S1. Centered around the approach of full-attention data augmentation, this paper establishes a rigorous concept of effective mutual information. The effectiveness of full attention data augmentation is validated based on the concept of effective mutual information.

S2. The experimental evaluation was conducted on multiple datasets of different scales and shows the efficiency of proposed model.

S3. This paper derives insights for the possible direction of enhancing the efficiency of graph contrastive learning.

# Weak points:

W1. The novelty of this paper is limited.

W2. This paper lacks sufficient explanation regarding the selection process of the three noise matrices and fails to adequately prove their effectiveness across different datasets.

W3. This paper contains inconsistencies and lacks clarity in its context.

# Detailed comments:

1. This paper presents a single primary innovation, which is the introduction of noise matrices as a data augmentation technique in the full-attention transformer. Additionally, the combination of the full-attention transformer with graph contrastive learning has also been employed by other researchers in the field.
2. In Section 4.3, within the framework of effective mutual information, this paper verifies the theorem that a noise matrix {0, 1} can reduce the invalid mutual information brought by the full-attention transformer in graph contrastive learning. However, the effectiveness of the noise matrices {0, -1} and {1, -1} has not been verified.
3. In the comparative experiment in Section 5.3, three different noise matrices showed different performances on the NCI1 dataset. However, the paper did not provide a reasonable explanation for the poorer performance of the noise matrices {0, -1} and {1, -1}. Additionally, the performance of these three different noise matrices was only tested on one dataset. It would be advisable to include additional experiments on datasets with different levels of data sparsity to further investigate whether there are any differences in the performance of the three noise matrices.
4. Similarly, in the comparative experiments regarding the probability ω of the noise matrix in Section 5.3, experiments were only conducted on the NCI1 dataset. It is recommended to add additional experiments for validation.
5. In the results of the graph classification task shown in Table 1, there is no explanation provided for the significance represented by the numbers highlighted in red and blue. It is recommended to provide explanatory notes for clarification.
6. In Section 5.3, there are several misattributions of figures in the analysis of experimental results. For example, "The results of this comparison are depicted in Figure 5" actually corresponds to Figure 4.

# Edit:

Thanks to the authors for the clarifications. I would suggest that these clarifications be included in the paper to make it easier to understand. Based on the author's clarifications, I will make changes to the novelty.

**Questions:**

1. In the comparative experiments of noise matrices in Section 5.3, regarding the probability of noise matrices and the three different types of noise matrices, the experiments were only conducted on one dataset. Would the results vary due to differences in dataset sparsity?
2. This submission verifies that invalid mutual information between two views can be reduced by attending to a noise matrix, while valid mutual information is also reduced. The use of InfoNCE loss allows for the recovery of the decreased valid mutual information. How can we verify that the decreased valid mutual information is less than the recovered valid mutual information?

**Reviewer Confidence:**

3: The reviewer is confident but not certain that the evaluation is correct

**Scope:**

3: The work is somewhat relevant to the Web and to the track, and is of narrow interest to a sub-community

---

### Decision · Program_Chairs · 2024-01-22

**Decision:**

Accept

**Comment:**

This submission introduces a 'Full-Attention Graph Transformer' method for generating effective contrastive pairs in machine learning models. By expanding the encoder's latent space, this approach effectively mitigates issues like noisy data and ineffective data augmentations. The core idea is to utilize a 'Noise Matrix' combined with the full-attention matrix. Furthermore, the paper defines 'Effective Mutual Information' and provides several theoretical results to substantiate the efficacy of the proposed method. Empirical demonstrations are performed on several small datasets with interesting results.

 In terms of novelty, it collects ratings of 4, 4, 4, 5 averaging at 5.25. Regarding technical quality, the ratings are more varied with scores of 4, 3, 6, 4, that is, an average of 4.25. Overall, the reviewers are slightly in favor of this work. In addition, the paper's relevance is aligned with the graph track and the Web Conf in general.

 Strengths: Most reviewers like the idea of full-attention data augmentation and effective mutual information, and value the theoretical analysis provided. These form the main strength of this work.

 Areas of improvements: The concerns raised by the reviewers include 1) the unclear presentation with confusing notations and a lack of coherent organization, lack of a time complexity analysis, a relatively over-complicated method.

 Despite these concerns and issues, the submission does not face any insurmountable obstacles to publication. It could be a strong submission if the execution was better managed. Considering all these factors, the paper receives a positive recommendation from the AC.